# Use of a LHFB Device for Testing Mode III in a Composite Laminate

**DOI:** 10.3390/polym11081243

**Published:** 2019-07-26

**Authors:** Carlos Bertorello, Antonio Argüelles, Victoria Mollón, Jorge Bonhomme, Isabel Viña, Jaime Viña

**Affiliations:** 1Department of Materials Science and Metallurgical Engineering, University of Oviedo, 33203 Gijón, Spain; 2Department of Construction and Manufacturing Engineering, University of Oviedo, 33203 Gijón, Spain

**Keywords:** composite, fracture, fatigue, mode III

## Abstract

The present paper studies the fatigue delamination behaviour of an epoxy/carbon composite material under mode III loading using a longitudinal half fixed beam (LHFB) device initially designed for mode III static tests of composite materials formed by the stacking of plies. For this purpose, a series of tests was carried out at different levels of loading representative of the fatigue behaviour of the material, from the crack onset phase through the delamination phase to final fracture. The experimental results were treated statistically, obtaining the values of the fatigue limit for probabilities of fracture of 5% and 50%. Finally, a fractographic analysis of the fracture surfaces was performed which allowed us to identify the same characteristic patterns of static mode III fracture, namely broken fibres, cusps and saw-teeth, in addition to a new morphology consisting of the formation of agglomerations of resin produced by the friction between the lips of the specimen in the fracture zone that point to dynamic mode III fracture. These agglomerations eventually crack and become detached from the fibres, leaving these free of resin.

## 1. Introduction

From the moment that the manufacture of plastic matrix composite materials from the successive stacking of plies was implemented at an industrial level, it was evident to all those involved that a series of materials with very good mechanical features had been obtained, but that these materials presented a weak point. This weakness is located in the space between the different stacked plies. Regardless of the manufacturing process and no matter how high the matrix/fibre interaction is, the interlaminar space presents a certain weakness with respect to the rest of the material. Thus, the concept of delamination, i.e., the growth of cracks in the aforementioned space, arose in materials of this type. This delamination requires only a small defect, pore or discontinuity that acts as a stress concentrator and hence as an initiator in order for it to become a problem. It should be borne in mind that the problem of the generation of an interlaminar crack can be exacerbated when the growth of the crack is hidden, without being visible from the outside and without it being possible to take measures that slow down its advance until catastrophic failure of the part that contains it takes place, depending on the type of applied load.

Once the initiator exists, a specific stress state is required ahead of the crack for it to effectively progress. This is not necessarily a problem in any part that is in service and hence subject to load, provided that this possibility has been taken into account in its design. Depending on the type of stress state, pure fracture modes or all the possible combinations of these may be produced. Within pure modes, modes I and II have been extensively studied, both statically and dynamically [1,2,3,4,5]. However, mode III fracture has been analysed much less frequently. The main reason for this has been the difficulty posed by its simulation in the laboratory and, in addition, the not necessarily negligible mode II component that many of the designed devices contribute to the test. Of all the existing test methods, the most widely used and accepted by a qualified majority of researchers is the edge crack torsion (ECT) test that Liao and Sun used [6] for thick specimens, based on previous studies by Li and Wang [7]. Many research groups subsequently adopted this specimen geometry and the proposed loading mode [8,9,10,11]. In many cases, numerical methods of analysis have been used to prove that mode III fracture is much more common compared to mode II fracture in this type of test specimen geometry and for this test methodology [12,13,14,15,16].

Nonetheless, the ECT device and its successive derivations have not been shown to be valid for analysing the mode III behaviour of materials of this type when they are subjected to dynamic fatigue loading. In this case, as demonstrated in this paper, only the longitudinal half fixed beam (LHFB) device [17,18] allows the fatigue behaviour of composite laminates to be studied in a relatively simple way.

## 2. Experimental Procedure 

The designed device [17] was used in this study to test specimens with a typical double cantilever beam (DCB) geometry manufactured from a composite laminate. This laminate was manufactured using a MTM45-1/IM7 (12k)—134 g/m^2^ prepreg with 32% by weight of resin. MTM45-1 is an epoxy resin with a flexible curing temperature and high mechanical and toughness properties that can be used in infusion systems and for curing in a vacuum bag or in an autoclave. IM7 carbon fibre, on the other hand, is a high-strength continuous fibre with high deformation properties obtained from polyacrylonitrile (PAN). The surface of this fibre was treated in order to improve its adherence to the matrix.

The dynamic load was applied at a distance of 1 mm from the crack front, given that the existing percentage of mode II fracture is negligible compared to mode III fracture in this situation, as previously demonstrated [18].

The LHFB device was coupled to a Walter + Bai axial-torsional servohydraulic machine (Figure 1, Löhningen, Switzerland) that allowed the application of a sinusoidal torsional stress in such a way that the maximum and minimum values of the applied torque were kept constant throughout the test, with a ratio between them (asymmetry coefficient) of 0.1. This torque was always applied in the same direction, generating an initial maximum angle of value θ that increased slightly as the number of cycles increased (Figure 2).

The configuration of the laminates was symmetrical with a unidirectional alignment of the fibres, using a 15 mm thick ply of Tygavac RF-260-R in the mid-plane as an insert that acted as the initiator of delamination. The dimensions of the DCB specimen used in this study were: length = 200 mm, width = 10 mm and thickness = 6 mm.

The aim of the fatigue experimental program was to determine the fatigue curves of the material tested, when they are subjected to delamination processes with fracture under mode III, and dynamic stress. In this work it has been considered that fatigue failure has occurred in the element when the tested sample breaks.

A total of 23 tests were carried out; these tests were carried out at constant ΔG stress levels (14, 16, 21, 34, 45 and 53) combined with single tests. For its definition, the results obtained from the previous characterization of the material at static level were taken as a reference, calculating these levels as percentages of the critical energy release rate *G*_c_. 

## 3. Results

Tests were carried out with different torques applied on the mobile arm, using the Timoshenko beam theory to obtain the values of the energy release rate.

According to this theory, the total displacement (δ) will be:(1)$$\delta_{T}=\;\delta_{flexion}+\;\delta_{cortante}=\;\frac{PL^{3}}{3EI_{z}}+\;\frac{3PL}{2bhG\prime }$$
and the energy release rate (*G*) will be:(2)$$G=\;\frac{1}{b}(\frac{P^{2}L^{2}}{2EI_{z}}+\;\frac{3P^{2}}{4bhG\prime })$$
where *P*: applied load*E*: Young’s modulus*G′*: Shear modulus*I*_z_: moment of inertiaL: length of the specimenb: width of the specimenh: thickness of the specimen.

Figure 3 which shows the results for all the specimens, was obtained by applying this formulation to the test specimens.

Fatigue failure was considered to occur when complete delamination was produced. Under this fracture mode, the process of growth of the fracture phenomenon was unstable and occurred almost simultaneously with the crack initiation process.

In order to achieve a suitable interpretation of the fatigue behaviour of a composite material, it is advisable to perform a statistical analysis of the sample obtained, given its relatively high dispersion. In this study, we used a model based on a Weibull distribution proposed by Castillo et al. [19] which has already been successfully used in this type of material, though applied to another fracture mode [20]. This model allows the fatigue life to be obtained from a representative sample of experimental data.
(3)$$P_{f}=F\left(N;G\right)=1-exp\left[\left(\frac{log\left(N/N_{0}\right)\cdot log(G_{max}/G_{0})-\lambda }{\delta }\right)\right]\;$$
with
(4)$$log\left(N/N_{0}\right)\cdot log(G_{max}/G_{0})\geq \lambda$$
where *N* is the fatigue life measured in cycles; *G*, the applied range of loading; *P*_f_, the probability of failure; and *G*_0_, *N*_0_, β, λ and δ are the parameters to be estimated, with the following denotations:*N*_0_: threshold value or limit number of cycles*G*_0_: limit value of the ERR (Energy Release Rate)β: shape parameter of the Weibull distributionλ: the Weibull location parameter determining the position of the limit curve associated with the zero probability of failureδ: scale parameter.

Figure 3 shows the percentile curves for the tested material that correspond to the probabilities of fatigue fracture of 5% and 50%, according to the Weibull model, plotting the maximum fracture energy versus the number of cycles required for the onset of the fatigue crack. The typical shape of the fatigue curves can be appreciated. It can likewise be seen that the fatigue limit for 50% may be estimated around 210 J/m^2^, and that for 5%, around 190 J/m^2^.

As soon as the suitability of the model used for the prediction of the behaviour of the material was verified a good adjustment of the experimental data with the results derived from the model, particularly for low probabilities of fracture, only one of the experimental results modified this tendency. Another important aspect to highlight was the low capacity of the material to withstand loads under mode III fatigue stress, given that there was complete delamination of the material for fracture energies significantly lower than those obtained under the static regime and for a relatively low number of fatigue cycles. This indicates the low capacity of the material to withstand this mode of stress under the dynamic regime both in the low number of cycles zone and in the theoretical infinite life zone; for the fatigue tests performed, one million cycles were considered as the fatigue limit.

Figure 4 presents the amplitude of the energy relaxation rate derived from the experimental data obtained, expressed as the percentage of the delamination energy obtained in the previous static characterisation of the material, which could be applied to an element subjected to fracture under mode III fracture and fatigue load, and has been considered as stress amplitudes: Δ*G* = *G*_max_ − *G*_min_, versus its fatigue life for a 5% probability of failure.

The relatively high slope of the adjustment line of the experimental data obtained indicated a marked loss of fatigue properties as the level of stress applied to the samples tested decreased, and in the zone with a high number of cycles, stress levels between 27% and 32 % were also observed. The high dispersion of the results obtained shows that this behaviour of the material is justified by the failure mode that occurs under this type of fracture stress in which the fatigue fracture of the sample tested occurs unsteadily, generally instantaneously when the initiation of a crack to fatigue occurs, i.e., without an appreciable growth phase. This behaviour is slightly modified for low levels of stress in which, after the crack initiation, there is a more or less brief period of crack growth in which the appearance of small fibre bridges is possible and which increases the number of cycles of the samples tested at low levels of stress and, consequently, increases the dispersion of the results obtained.

## 4. Fractography

A fractographic study was carried out on the fracture surface of the specimens that had previously been tested using the LHFB device, employing a JEOL 6610LV scanning electron microscope (SEM, Akishima, Japan) for this purpose.

The analysis was performed both in the region close to the insert and in more internal areas with the aim of finding typical features of mode III fracture [20,21], as well as particular features corresponding to fatigue testing under this loading mode.

Figure 5 shows a fracture plane in a region near the insert. In the most enlarged images, both the saw-teeth [21] and the cusps [21,22] characteristic of this fracture mode can be observed under static loading.

Figure 6 shows another typical feature of this type of fracture, namely broken fibres, as well as cusps. In the specimens tested at low load levels, the generation of fibre bridges has been mentioned above. In this micrograph, obtained from a specimen tested under these conditions, the presence of broken fibres from fibre bridges can be seen.

Finally, a novel finding in this study was the appearance, both in regions close to the insert and in those at a distance, of agglomerations of “beds” of resin (Figure 7) created by friction resulting from the effect of fatigue. These beds subsequently crack and become detached, leaving the areas of fibres that can be seen in the upper left part of the photograph free and visible.

## 5. Conclusions

The LHFB device has been shown to be perfectly valid for performing mode III fatigue tests on composite laminates.

The statistical regression model based on a Weibull distribution used to evaluate the experimental results allows additional information to the results of the fatigue test to be obtained, such as the estimation of fatigue strength for different loads to those employed in the test. It was also possible to determine a fatigue limit value of 25 J/m^2^ for a 5% probability of fracture and 45 J/m^2^ for a 50% probability of fracture, a fundamental parameter in fatigue behaviour that is always difficult to obtain.

Fractographic analysis confirmed the existence of the typical morphology in dynamic mode III fracture, namely the existence of broken fibres, cusps and saw-teeth. Furthermore, a novel fractography was observed, consisting of the presence of substantial accumulations of resin, denominated “beds”, that arise due to the friction occurring in this type of fatigue between the lips of the crack as it is displaced.

## Figures and Tables

**Figure 1 polymers-11-01243-f001:**
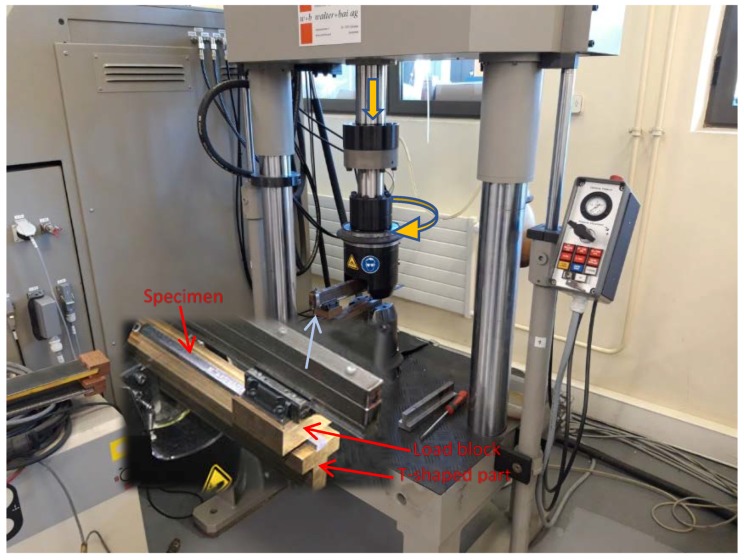
Testing equipment.

**Figure 2 polymers-11-01243-f002:**
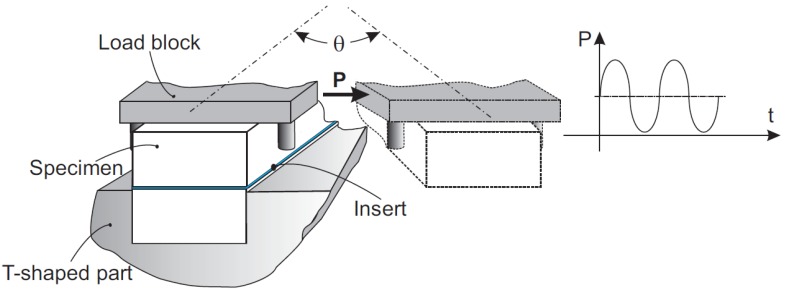
Detail of the method for applying the load.

**Figure 3 polymers-11-01243-f003:**
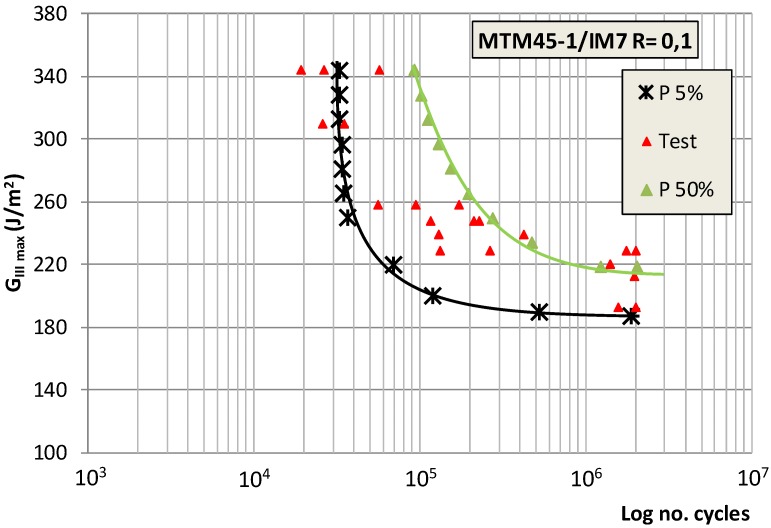
Fatigue behaviour of the material for fracture probabilities of 5% and 50%.

**Figure 4 polymers-11-01243-f004:**
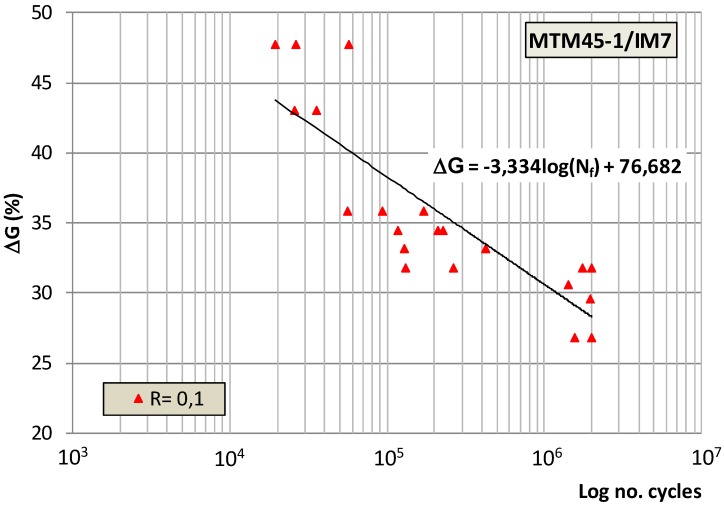
Fatigue loading levels expressed as a percentage of the static energy release rate.

**Figure 5 polymers-11-01243-f005:**
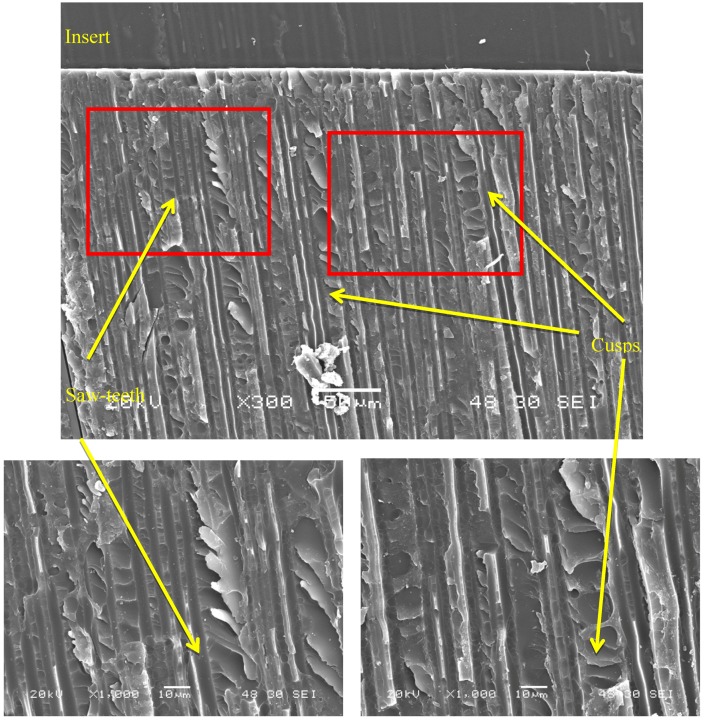
Presence of saw-teeth and cusps in a region near the insert.

**Figure 6 polymers-11-01243-f006:**
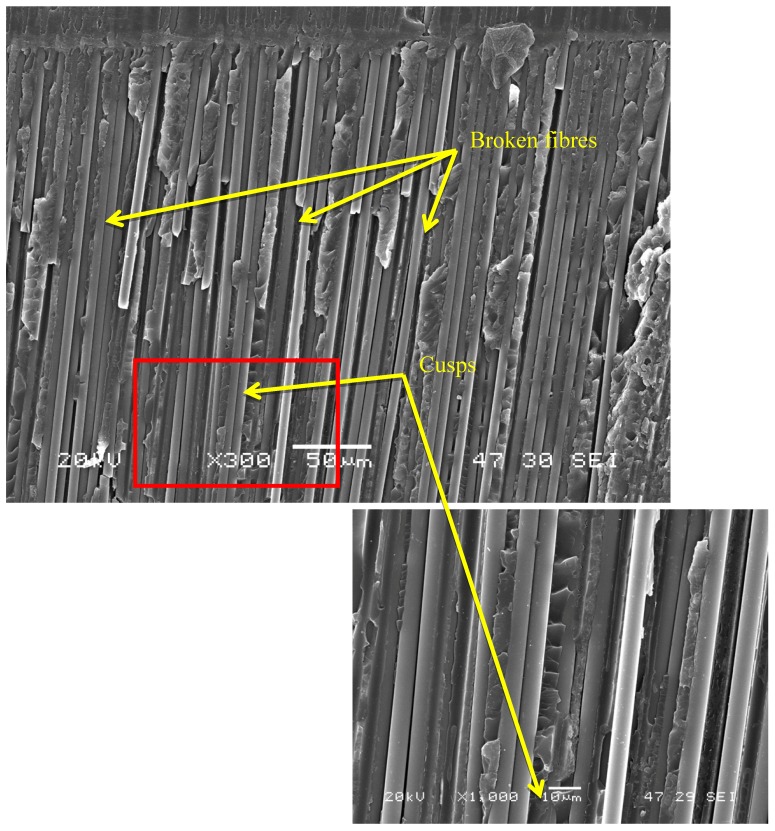
Presence of broken fibres and cusps in a region near the insert.

**Figure 7 polymers-11-01243-f007:**
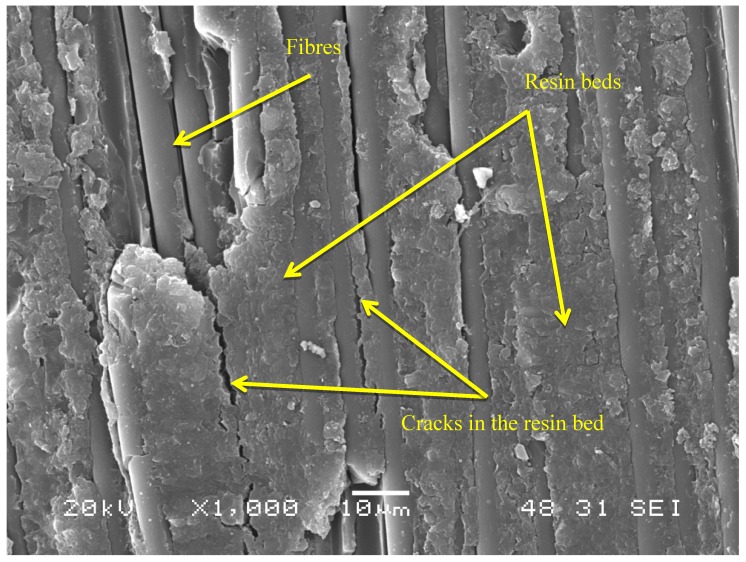
Presence of resin beds.

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
