# Peer review of "Use of a LHFB Device for Testing Mode III in a Composite Laminate"

_polymers, 2019, doi:10.3390/polym11081243_

Round 1
Reviewer 1 Report
This paper presents the experimental data of mode III test on composite samples along with a fractography analysis to interpret the failure mechanisms.
The publication has technical merit and it is worth publishing, however some spelling mistakes and readjustments are needed to comply with the journal standards.
The author is invited to change figure 1 ass it does not lead the reader to visualise the mode II test jigs properly. It would be good to provide a schematic draw of the fixture and the real one employed.
How many samples were tested?
I would suggest the author to revised the manuscript giving more details of the fractography analysis linked with the experimental data of the fatigue test.
I would be willing to review this paper again after this minor revision
Author Response
1. The author is invited to change figure 1 ass it does not lead the reader to visualize the mode III test jigs properly, It would be good to provide a schematic draw of the fixture and the real one employed
Response: Figure 1 has been completed in order to detail the testing equipment.
2. How many samples were tested?
Response: It has been answered in the text.
3. I would suggest the author to revise the manuscript giving more details of the fractography analysis linked with the experimental date of the fatigue test.
Response: It has been made.
Reviewer 2 Report
The paper seems to be of high quality, it follows long therm studies and publication of the authors.
Suggestions:
The results of the study should be more detailed confronted with other authors.
Figure 1 - description of the testing equipment should be more detailed (for the readers who do not read the full paper).
Equation 1 and 2, please describe all the quantities.
Please discuss more results in figures 3 a 4.
More discussion with other authors is needed.
Author Response
1. The results of the study should be more detailed confronted with other authors
Response: We agree, the problem is that, in the exhaustive bibliographic search carried out, no study of fatigue in mode III fracture has been found.
2. Figure 1- description of the testing equipment should be more detailed (for the readers who do not read the full paper)
Response: Figure 1 has been completed in order to detail the testing equipment.
3. Equation 1 and 2, please describe all the quantities.
Response: It has been made.
4. Please discuss more results in figure 3 and 4.
Response: It has been made.
More discussion with other authors is needed.
As previously replied, it has been impossible for us because there is no bibliography on the subject.